# Fine Tuning: Effects of Post-Translational Modification on Hsp70 Chaperones

**DOI:** 10.3390/ijms20174207

**Published:** 2019-08-28

**Authors:** Alijah A. Griffith, William Holmes

**Affiliations:** Rhode Island College, Biology Department, Providence, RI 02908, USA

**Keywords:** Hsp70, post-translational modifications, chaperone proteins

## Abstract

The discovery of heat shock proteins shaped our view of protein folding in the cell. Since their initial discovery, chaperone proteins were identified in all domains of life, demonstrating their vital and conserved functional roles in protein homeostasis. Chaperone proteins maintain proper protein folding in the cell by utilizing a variety of distinct, characteristic mechanisms to prevent aberrant intermolecular interactions, prevent protein aggregation, and lower entropic costs to allow for protein refolding. Continued study has found that chaperones may exhibit alternative functions, including maintaining protein folding during endoplasmic reticulum (ER) import and chaperone-mediated degradation, among others. Alternative chaperone functions are frequently controlled by post-translational modification, in which a given chaperone can switch between functions through covalent modification. This review will focus on the Hsp70 class chaperones and their Hsp40 co-chaperones, specifically highlighting the importance of post-translational control of chaperones. These modifications may serve as a target for therapeutic intervention in the treatment of diseases of protein misfolding and aggregation.

## 1. Post-Translational Modifications

All proteins consist of specific combinations of twenty different amino acids. Since the residues of amino acid side chains differ in size, charge, and polarity, each amino acid offers an incredible amount of biochemical diversity that defines the subsequent structure and function of each respective protein after translation and folding. While the amino acid sequence of a protein cannot be altered post-translationally, the chemistry of specific amino side chains can be altered through a collection of post-translational modifications (PTMs). Modifying the biochemical characteristics of a given protein through PTMs immediately results in the modified protein exhibiting an altered form, function, and dynamic, exponentially increasing proteomic diversity within the cell and allowing for the cell to respond efficiently to specific stimuli.

PTMs occur in response to changing cellular environments and are catalyzed on specific residues of target proteins by a collection of specialized enzymes. Typically, these enzymes are part of a greater network of cell communication pathways to allow for the immediate response to altered cellular conditions. The ability to enact an immediate cellular response to dynamic environmental conditions is generally advantageous compared to the synthesis of proteins de novo, as the latter is associated with greater metabolic expense. Through PTMs, proteins can be synthesized in one form and subsequently modified and perform alternative functions upon stimuli. For example, glycogen phosphorylase is an enzyme that catalyzes the breakdown of glycogen to produce glucose. Glycogen phosphorylase activity can be attuned to the needs of the cell: if there is sufficient energy, the enzyme remains inactive; upon low energy signaling, cascades will lead to the phosphorylation of glycogen phosphorylase to activate glucose production [1]. Modification may also lead to changes in subcellular localization and protein turnover, as observed in the case of ubiquitin.

Several types of PTMs, including phosphorylation, methylation, and acetylation, occur through the addition of small functional groups (e.g., phosphate, methyl or acetyl functional groups, respectively) to specific side chains within the target protein. Phosphorylation occurs at the terminal hydroxyl residues of serine, threonine, and tyrosine, and significantly alters the charge of the protein by adding a –2 charge to the modified residue [2]. Methylation and acetylation typically occurs at lysine residues, acting to neutralize the residue while adding large steric constraints [3,4]. Larger macromolecules such as carbohydrates and lipids may also be used to modify proteins. Carbohydrate chains are added to proteins within the Golgi body and are typically added to improve the specificity of receptor proteins [5]. Fatty acid chains are also added to proteins to act as a lipid anchor, tethering proteins to organellar membranes or the plasma membrane [6]. Lastly, proteins are also covalently modified with other proteins via covalent modification on lysine residues; the most well-known are ubiquitin and the ubiquitin-like proteins [7].

While it is known that a dynamic assembly of cellular proteins undergo PTMs, there is mounting evidence that chaperone proteins are also subject to such modification. In brief, chaperone proteins are ubiquitous, conserved enzymes that maintain proteostasis within the cell by mediating the protein structure in vivo via a variety of mechanisms, acting as a preventative measure against stress-induced protein misfolding and aggregation [8]. The limited number of known chaperone-encoding genes are at odds with the growing collection of interactions and functions that chaperones are known to perform. Recent proteomic work has identified several PTMs in chaperone proteins, suggesting a potential mechanism for the functional pleiotropy observed for several classes of chaperones. Controlling chaperone function via PTMs likely circumvents the energetic cost of additional protein synthesis needed to otherwise compensate for such functional diversity. While it is known that several classes of chaperone proteins are likely subjected to PTMs, one family of particular interest concerning this topic is that of the Hsp70 class chaperones. Hsp70 chaperones have a high degree of conservation across all domains of life and are implicated in human pathologies. As a result, this review will focus on the post-translational modification of Hsp70 class chaperones as well as their interacting co-chaperones.

## 2. Hsp70 Structure and Allosteric Regulation

Hsp70 proteins are dynamic chaperones notable for their functional diversity, which includes both promiscuous association to either unfolded, misfolded, or aggregated substrates as well as selective binding to folded substrates [9,10,11]. Hsp70 is structurally composed of two domains: an N-terminal 44 kDa ATPase nucleotide binding domain (NBD) and an 18 kDa C-terminal substrate-binding domain (SBD), which interacts with hydrophobic regions of client proteins (Figure 1) [12,13,14,15,16]. The NBD contains four subdomains that are further separated into two major lobes (I and II) by a central ATP/ADP binding pocket. The SBD contains a two-layered β-sandwich domain (SBDβ) that encompasses the substrate binding site, a 10 kDa α-helical lid domain (SBDα), which shields the substrate binding site, and a C-terminal intrinsically disordered region (C-IDR) [17,18]. Notably, the versatility of Hsp70 activity is attributable to the degenerative recognition motif located within the substrate binding site of the SBDβ, which is composed of five core hydrophobic amino acids situated between positively charged residues and is advantageous for client binding [19]. The NBD and SBD domains are connected by a highly conserved interdomain hydrophobic linker (L_L,1_) that allows the allosteric coupling of the domains, which is indispensable for proper Hsp70 function [10,18]. The intricate mechanism by which Hsp70 facilitates the refolding of client proteins is heavily dependent on interdomain allosteric communication and is regulated by nucleotide binding, substrate identity, Hsp40/DnaJ-family co-chaperones, and the association of nucleotide exchange factors (NEFs) [15,16,20]. Hsp40/DnaJ-family co-chaperones promote and coordinate the intrinsic ATPase activity of Hsp70 with substrate binding, while ADP/ATP exchange is catalyzed by the GrpE, Bag1, Fes1/HspBP1, and Sse1/Hsp110 families of NEFs [16]. The mechanism of Hsp70 activity is a repetitive cycling between two conformational states, as dictated by ATP hydrolysis, which is stimulated with the association of both the substrate peptide and J-domain co-chaperones to Hsp70 [9,15]. While ATP is bound, Hsp70 assumes a conformation that exhibits a low affinity for substrates accompanied with high substrate association and dissociation rates [15,21,22,23] In turn, ATP hydrolysis prompts a conformation that displays a 100- and 1000-fold decrease in substrate association and dissociation, respectively, and a 10- to 50-fold increase in substrate affinity [15,21]. Reiteration of this conformational cycle results in the promotion of the client protein to its native structure [21]. The regulatory mechanisms inherent to Hsp70 allostery remain unknown.

Members of the Hsp70 chaperone family exhibit either constitutive gene expression or are stress inducible, hence their classification as heat shock proteins [24,25,26]. Genomic sequencing determined that eukaryotes express distinctive isoforms of Hsp70, which exhibit characteristic localization to a vast distribution of intracellular compartments [25,27]. Cognate isoforms of Hsp70 are present in all major intracellular compartments including the cytoplasm, mitochondria, or endoplasmic reticulum. Inducible Hsp70 isoforms primarily exhibit a cytoplasmic or nuclear distribution [24,25,27,28].

Organelle-associated Hsp70s are typically encoded by single genes in the majority of organisms, although cytosolic Hsp70 isoforms are encoded by a series of homologous genes [24]. In *Saccharomyces cerevisiae* (yeast), there are at least fourteen Hsp70 isoforms that are categorized into five subclasses according to conservation of amino acid sequence identity [29,30]. Only two subclasses, Ssa and Ssb, are known to occupy the cytoplasm [27]. In contrast, the single-membered Ssc (Ssc1) and Ssz (Ssz1) subclasses are localized to the mitochondria, while the Ssd (Ssd1/Kar2) subclass is associated with lumen of the endoplasmic reticulum [27,31].

The Ssa (stress-seventy A) Hsp70 subfamily consists of four members, Ssa1–4, which are highly homologous but differ in their expression patterns [32]. Of the cytosolic Hsp70s, only the Ssa subclass is considered essential [26,33]. Ssa1/2 are constitutively expressed, with basal levels of Ssa2 exceeding that of Ssa1 under normal growth conditions; however, Ssa1 expression is induced via cellular stress to levels equivalent to that observed for normal Ssa2 expression [26,32]. *Ssa3* and *Ssa4* genes are strictly stress-inducible, with *Ssa3* being induced under starvation conditions and Ssa4 induced primarily by thermal stress [30,32]. While simultaneous disruption of all four Ssa genes is lethal, there is indication that there is some degree of functional redundancy among the Ssa subfamily, as the overexpression of one member is sufficient for cell viability [26]. Acknowledging the possibility of functional redundancy among Ssa members, it is notable that *ssa1ssa2* double mutants exhibit slowed growth and thermosensitivity, suggesting that complete complementation among the Ssa genes is not possible, even with overexpression [25].

Like the Ssa Hsp70s, the Ssb subclass of Hsp70 chaperones also occupies the cytoplasm [34,35]. The Ssb Hsp70 subclass is comprised of two genes, *Ssb1* and *Ssb2*, neither of which are inducible by heat shock [35]. Within the cytoplasm, Ssb1/2 are predominantly associated with translating ribosomes and are thought to form stable complexes with nascent polypeptide chains, to the extent that it was proposed that Ssb1/2 act as core components to translation machinery [36]. The Ssb subfamily is not essential, but disruption of Ssb1/2 manifests in slowed growth at all temperatures, cold sensitivity, and hypersensitivity to translation inhibitors [29,34].

While the Ssa and Ssb Hsp70 subfamilies share 60% amino acid identity and are both cytosolic in distribution, members of these subclasses are unable to functionally substitute for one another, even when Ssa1 is expressed from the Ssb1 promoter or vice versa [27]. The functional distinction between Ssa and Ssb members was hypothesized to be the result of differences in substrate specificity because of the dissimilar recognition motifs within the variable domains of Ssa1 and Ssb1 [37]. This hypothesis was shown to be unlikely, as it was determined that expression of a chimeric Hsp70 protein, containing the variable domain of Ssa1 but the peptide-binding domain and ATPase domains of Ssb1, could sufficiently rescue Ssb-specific phenotypes in *ssb1ssb2* deletion mutants [37].

This is not to argue that there is no functional distinction between cytosolic, endoplasmic reticulum, or mitochondrial Hsp70s, as such distinctions have been thoroughly documented [38,39]. Rather, such functional distinctions between Hsp70 isoforms are more likely attributable to specific interactions between Hsp70 and their respective co-chaperones.

## 3. Hsp70 Co-Chaperones

Members of the Hsp70 family act in concert with individual Hsp40 co-chaperones to facilitate a variety of cellular processes via a highly conserved, transient mechanism [40,41,42]. In brief, Hsp40s target Hsp70s to specific substrates by stabilizing Hsp70 substrate complexes and promoting Hsp70 ATPase activity [9,43]. Hsp40 co-chaperone activity is thought to be essential to Hsp70 chaperone efficiency, although the mechanism by which Hsp40s bind and transfer non-native substrates to Hsp70 is not yet known [42,44].

Hsp40 family members exhibit distinct combinations of four conserved domains initially described in *Escherichia coli* DnaJ: the J-domain, a G/F-rich containing region or G/F-rich motif, a zinc-finger-like domain, and a C-terminal peptide binding fragment [45,46]. Only certain Hsp40s display conservation across the four established domains; thus, the Hsp40 family is categorized into three subtypes according to domain composition [44,47,48]. Only type I Hsp40s are known to share all four canonical domains with the proteotypic DnaJ [44].

All Hsp40 family members contain a J-domain, a helical hairpin structure of 70–75 residues in length [9,44,46]. The J-domain is the most highly conserved of all domains characteristic to Hsp40s, as it can exhibit amino acid sequence identity ~50% across Hsp40 family members, and it has been used to characterize additional DnaJ homologs since its initial observation [9,47,49]. The J-domain is present within N-termini of type I Hsp40s, including Ydj1 in yeast and Hdj2 in humans, and type II Hsp40s, such as Sis1 in yeast and Hdj1 in humans [42,50]. The J-domain contains the family signature HPD motif responsible for Hsp70/Hsp40 association and Hsp70 substrate binding by regulating the hydrolytic cycle of Hsp70 activity [42,51,52]. In type III Hsp40s, the J-domain is not exclusive to any particular location and is the only domain characteristic to Hsp40s that is present. Type III Hsp40s otherwise contain a vast variety of other domains and motifs [44,50].

Following the J-domain is the G/F-rich motif, a stretch of approximately 30 residues that are rich in the amino acids glycine and phenylalanine [45,49]. Only present in type I and II Hsp40s, the G/F-rich motif acts as a flexible hinge connecting the J-domain to the C-terminal peptide binding fragment and is necessary for activating the substrate binding activity of Hsp70 [44,49,52,53]. The G/F-rich motif is present immediately downstream of the J-domain and exhibits notable conservation, including a nearly universally conserved Asp-Ile-Phe (DIF) motif, suggesting its importance for Hsp40 function. However, considering type III Hsp40 members do not contain this motif, it was suggested that the G/F-rich motif is nonessential for certain subsets of Hsp40 chaperone functions [49]. Type I Hsp40s additionally contain a zinc-finger-like region (ZFLR) and conserved C-terminal domains I and II (CTDI and CTDII), which are located downstream from both the J-domain and G/F-rich motif and are required for type I Hsp40s to bind and interact with folding intermediates and for Hsp70 to interact with non-native polypeptides [42,50,52,54]. Type II Hsp40s contain all other aforementioned domains but lack the ZFLR, which is instead replaced by a region rich in glycine and methionine residues [42,52,54]. Type I and type II Hsp40s are not structurally or functionally redundant; however, both can act as co-chaperones and recruit non-native polypeptides to Hsp70 through an unknown mechanism. Type III Hsp40s fail to bind to non-native substrates [46,52].

Notably, variations in domain composition among Hsp40 members are thought to be responsible for differences in substrate selectivity and functional diversity among both individual Hsp40 proteins and Hsp70/Hsp40 functional pairs [9,41,51]. For example, it was determined that interaction between Hsp70 and its respective type I or type II Hsp40 co-chaperones differentially affected Hsp70 chaperone function. While the mechanism by which type I and type II Hsp40s alter Hsp70 activity remains to be understood, Hsp70 members are often found to be colocalized with members of the Hsp40 family. Presumably, this colocalization allows for the generation of specific Hsp70/Hsp40 functional pairs that are capable of performing highly specialized processes in a specific manner [55,56]. It is reasonable to assume that the action of said functional chaperone/cochaperone pairs can be modulated by PTMs (Figure 1 and Table 1) [55,56].

## 4. Modulating Chaperone Activity through Post-Translational Modification

### 4.1. Phosphorylation

Proteomic meta-analysis identified regions of Hsp70 that are enriched for phosphorylation. Many phosphorylation sites cluster at the N-terminal NBD (S19, S23, S25) and at the C-terminal SBD (T492, S495, T499) [57]. Alanine mutations in the N-terminal domain resulted in a modest growth rate in yeast, which was further exacerbated with C-terminal mutations. Phosphorylation at the N-terminal NBD favors client binding and aggregate prevention, whereas phosphorylation of the C-terminal SBD results in an inability to survive heat shock and decreased interaction with translating ribosomes [58]. The following studies highlight how Hsp70s are able to be fine-tuned via modification based on the immediate needs of the cell.

Hsp70 activity is required during times of growth due to rapid translation and protein accumulation. In this way, Hsp70 acts as a checkpoint for transitions in the cell cycle. Hsp70 acts to stabilize various cyclin proteins as the cell progresses though G1 to the S phase. In times of nutrient starvation, the kinase Pho85 phosphorylates Hsp70 at T38, which disrupts Hsp70 binding with the yeast Hsp40 co-chaperone, Ydj1. Loss of co-chaperone binding decreases chaperone activity and affinity for cyclin proteins. As a result, the cyclins are broken down, halting the cell cycle in G1 until nutrient conditions improve [59].

Hsp70 can also act as a checkpoint during mitosis. Plk1 is a kinase that is activated upon oxidative stress and is known to phosphorylate Hsp70 at multiple locations (T13, S362, T226, S631, and S633). Hsp70 is normally localized to the cytoplasm but will co-localize with the centrosome upon Plk1 activation and phosphorylation. Upon its localization, Hsp70 will stabilize centrosomes and spindle microtubules, halting mitosis until the oxidative stress is alleviated [60]. Additionally, Hsp70 is also phosphorylated by the kinase Nek6 at T66, causing Hsp70 to also localize to kinetochore microtubules. Presumably, Hsp70 stabilizes kinetochore complexes to slow mitosis as T66A mutants fail to localize to microtubules and decrease the rate of mitosis [61].

The Hsp40 co-chaperone HOP/STI1 is also phosphorylated in response to the cell cycle. When Sti1 is incubated with the cell cycle kinase casein kinase 2 (CK2) in vitro, a phosphate group is added to Sti1 at S189. Additionally, Cdc2 kinase also phosphorylates Sti1 at T198. It remains unknown if phosphorylation alters Hsp70 or Hsp90 binding, although the phosphorylation of HOP alters its localization from the cytoplasm into the nucleus and acts as a recruiting agent specifically for Hsp90 [62].

Hsp70 sits at a vital nexus to determine the fate of client proteins. Besides its refolding activity, Hsp70 can act as an agent for the ubiquitin-proteasome system and aid in tagging proteins for degradation. It is clear that phosphorylation (and acetylation) act as the molecular switches between these pathways (Figure 2).

Since Hsp70 interacts with a host of co-chaperones that define its overall function, it is of no surprise that PTMs can alter this dynamic. Hsp70 is typically found as a monomer when present in the cytosol; however, it was recently demonstrated that mammalian Hsp70 can form a dimer in the cytoplasm. Phosphorylation on T504 promotes and stabilizes antiparallel homodimers of Hsp70 in a structure that is reminiscent of the Hsp70-Hsp90 client loading complex. Phosphorylation and subsequent dimerization stabilizes client proteins and increases time to client release [63]. It is suggested that this could be a potential mechanism for proteins that require a more complex folding pathway to assume their native conformations. Questions remain about the physiological relevance of Hsp70 dimers, including the kinase responsible for this modification and the cellular signaling events that induce phosphorylation. It will be interesting to see if client binding is required for phosphorylation at T504 and how this might disrupt the typical bind-and-release mechanism of Hsp70.

There are patterns emerging that suggest how Hsp70 favors refolding client proteins over degradation and how HOP/Sti1 binding shifts the balance towards client refolding. Multiple phosphorylation sites play a role in stabilizing the Hsp70/HOP interaction and, thus, influence client refolding. The extreme C-terminus of Hsp70 is phosphorylated by casein kinase 1, casein kinase 2, and GSK3-β, and in vitro analysis of these phosphorylation events promotes binding to HOP. Phosphorylation of T636 increases Hsp70 affinity for HOP but does not appear to influence affinity of other chaperones. T636 is commonly found in excess in cancer cells to promote protein stability and favor a more active cell cycle [64,65]. Stabilization of Hsp70/HOP has also been observed using phosphomimetic mutants of Hsp70 and increases growth rate of cells [66].

CHIP (C-terminus of Hsc70 interacting protein) is an E3 ubiquitin ligase responsible for recognizing target proteins that will be modified with ubiquitin and subsequently degraded by the proteasome. CHIP behaves similarly to other co-chaperones binding to the extreme C-terminus of Hsp70. Peptide binding analysis suggest that Hsp70 has high affinity for HOP, CHIP, and DnaJ, but when the C terminal peptide of Hsp70 is phosphorylated, HOP and DnaJ are unaffected, yet CHIP losses all affinity [65]. C-terminal phosphorylation of Hsp70 favors HOP interaction and, thus, client protein refolding; while in its un phosphorylated state, Hsp70 favors CHIP binding and subsequent client protein degradation. CHIP itself is also phosphorylated via aurora kinase A, which inactivates its substrate recognition. Alanine substitutions within CHIP lead to a lack of phosphorylation, which results in target protein stabilization [67].

Beyond refolding or targeting client proteins for degradation, Hsp70 also influences apoptotic signaling pathways. In response to given environmental stressors, cells engage a variety of intracellular signaling pathways that promote either cell survival or initiate cell death [68]. Cell death is associated with two general mechanisms: programmed cell death and necrosis [69]. Programmed cell death encompasses both caspase-dependent (apoptosis) or caspase-independent pathways (e.g., autophagy, necroptosis, and apoptosis-like programmed cell death), which are activated in response to cellular insult and during normal aging, development, and morphogenesis [69,70]. Evidence indicates that heat shock proteins can modulate apoptotic signaling and regulate apoptosis by executing either a proapoptotic or antiapoptotic function [71]. Hsp70 is often reported to exhibit antiapoptotic activity. Elevated expression of Hsp70 was shown to significantly reduced apoptosis in mouse embryo fibroblasts treated with heat, tumor necrosis factor α (TNFα), and ceramide [72]. Notably, fibroblasts exposed to heat and TNFα exhibited reduced caspase activity, suggesting that Hsp70 may disrupt apoptosome formation through downregulating caspase activity [72]. This suggestion was further evidenced as later study determined that Hsp70 directly associates with the caspase-recruitment domain (CARD) of apoptosis protease activating factor-1 (Apaf-1), which in turn blocks the oligomerization of Apaf-1 and its association with procaspase-9, preventing caspase activation [73]. There is also evidence that Hsp70 affects caspase-independent mechanisms. It was determined that overexpression of Hsp70 in Apaf-1^–/–^ cells prevented cell death by interacting directly with the caspase-independent mitochondrial intermembrane flavoprotein apoptosis inducing factor (AIF), preventing AIF-induced chromatin condensation [71]. Considering apoptotic processes are essential for maintaining physiological homeostasis, it is possible that such pathways could be disrupted, including interruption of normal chaperone activity via its absence [74].

Apoptotic signaling can originate from internal or external stressors that ultimately involves a cleavage cascade of caspase proteins [70]. For example, cells treated with cigarette smoke extract (CSE) increase apoptosis by inhibiting Hsp70 expression and activity [75]. In human cell lines, the protein RAI16 acts as a scaffold between protein kinase A (PKA) and Hsp70; formation of this protein complex results in the phosphorylation of S486 on Hsp70. This modification stabilizes caspase-3, preventing its cleavage and the induction of apoptosis. Upon apoptotic signaling, RAI16 is phosphorylated, disrupting the PKA–Hsp70 scaffolded complex. Once Hsp70 is dissociated from PKA, it is dephosphorylated at S486 and exhibits a decreased affinity for caspase-3. Apoptosis is promoted once caspase-3 is no longer stabilized by Hsp70 [76]. Hsp70 is also influenced by lysine acetylation to alter apoptotic signaling.

### 4.2. Acetylation

Recent work has demonstrated that lysine acetylation also plays a role in modulating Hsp70 activity. We have already discussed the role of Hsp70 in apoptosis, but it is also involved in chaperone-mediated autophagy [77,78]. The process of autophagy refers to ‘cell eating’ wherein aggregated proteins, large portions of the cytoplasm, or even entire organelles are engulfed in a double membrane structure and digested by the lysosome [79]. Autophagy is induced during times of depleted nutrition, and the cell recycles its own macromolecules to meet energetic demand [80]. Under nutrient-rich conditions, Hsp70 interacts with Sirt2, a key metabolic regulator, and upon starvation, this interaction is disrupted [81]. Starvation induces K126 acetylation of Hsp70, increasing Hsp70′s affinity for the apoptotic regulator Bcl2 [82]. Upon Hsp70 binding, Bcl2 is inhibited, leading to the formation of the autophagosome. Acetylation at K126 decreases affinity for other Hsp70 co-chaperones and favors Hsp70 binding to Bcl2, promoting autophagy [83].

Apoptotic and autophagic signaling are also controlled by Hsp70 acetylation at K77. Overexpression of Hsp70 leads to a decrease in apoptotic signaling and appears to have a protective effect from lysosome membrane permeabilization. This protective effect is eliminated by a K77R mutation, indicating that K77 needs to be acetylated to inhibit apoptotic signaling [84]. It appears that this protective effect is the result of K77 acetylation favoring protein refolding pathways, as opposed to degradative or apoptotic pathways. Once K77 is acetylated, both autophagic- and apoptotic-inducing factors are stabilized, inhibiting these signaling pathways [85]. This suggests that K77 acetylation is key for promoting protein refolding, and acetylation increases during times of unfolded protein stress. Recently, it was demonstrated that the lysine acetyltransferase Ard1 becomes activated after cell stress and will acetylate Hsp70 at K77. Elevated rates of acetylation promote Hsp70 binding to HOP, favoring refolding, and increasing survivability during stress. However, under prolonged stress, Hsp70 is deacetylated at K77, which causes Hsp70 to functionally switch to favor CHIP binding and protein degradation rather than refolding [86].

While lysine acetylation is more widely studied, Hsp70 is also acetylated at the N-terminus by the N-terminal acetyltransferase A complex (NatA). N-terminal acetylation is a constitutive modification that happens co-translationally to all copies of Hsp70, as opposed to modifications that occur during a specific cellular response. Cytoplasmic proteins are acetylated in a sequence-dependent manner, and there are a range of effects on target proteins, from influencing binding partners to preventing degradation [87,88]. There is little known concerning the effects of N-terminal acetylation on Hsp70, although one report demonstrated that unacetylated Hsp70 exhibits increased binding to prion aggregates [89]. This suggests that N-terminal acetylation is required for optimal chaperone activity. Further study is required to determine the exact mechanism of N-terminal acetylation on Hsp70 activity.

### 4.3. Other Identified Post-Translational Modifications

Multiple reports of proteome-wide post-translational modifications offer insight to the extent of modification to which Hsp70 or its associated Hsp40s are subject. Hsp70 and Hsp40s were identified in multiple screens enriched for ubiquitinylation. These screens did not evaluate the physiological relevance of ubiquitinylation, nor did they describe the effect of these specific modifications. It remains to be determined if ubiquitnylation targets these chaperones for degradation or if it simply alters their subcellular localization. One report demonstrated how the co-chaperone Ydj1 delivers substrates to E3 ubiquitin ligases. The effect is not controlled by Ydj1 ubiquitinylation but presumably by interactions with Hsp70 [90,91,92].

A methylated form of Hsp70 was identified in human tissue culture cell lines. Jakobsson et al. identified a novel methyltransferase METTL21A, and they found that it interacts with Hsp70 isoforms in vivo. Additionally, METTL21A trimethylates K546 on Hsp70, and it is worthy to note that K546 is conserved in all Hsp70s, yet these organisms lack the modification or the requisite methyltransferase. Methylation at K546 does not alter the ATP hydrolysis rate, but it demonstrates a decreased binding to α-synuclein aggregates. The ability of methylated Hsp70 to prevent α-synuclein aggregate formation was also hindered [93].

Prenylation is the addition of an isoprenyl functional group to a protein. Isoprenyl groups are hydrophobic and typically localize proteins to a particular membrane, similar to a gycosylphosphatidylinositol (GPI) anchor [94]. Farnesylation is a subset of prenylation, where a farnesyl moiety is utilized to modify a protein. Hsp40 co-chaperone Ydj1 and its homologues are targets of farnesylation through a known recognition motif at their C-terminus. This modification appears to be constitutive, as the entire population of Ydj1 in the cell is farnesylated, and it appears that farnesylation is required to localize Ydj1 to the cytoplasmic face of the endoplasmic reticulum [95,96]. One interesting paradox is that all of the Ydj1 molecules are modified, and yet there is a mixed population of cytoplasmic and ER localized chaperone. Farnesylation is required for Ydj1 to aid in surviving heat stress and binding to amyloid aggregates, but it is not necessary for normal protein refolding [55,97].

Glycation is a post-translational modification where glycolysis intermediates are added to free amines of a protein. Glycation of proteins has gained interest recently as rates of glycation are increased in diabetics with high circulating glucose levels, potentially contributing to disease progression or age-related decline of protein function [98]. Methylglyoxal is a byproduct of glycolysis that can irreversibly modify proteins via a nonenzyme catalyzed reaction. Hsp70 was one of six proteins identified in a screen enriched for glycosylated proteins, along with a small heat shock protein Hsp26. It was found that Hsp70 was specifically glycated at K532, although this report did not investigate whether this modification resulted in any mechanistic changes [99]. However, it provides an interesting potential mechanism for age- and disease-related declines in protein folding stress within the cell. Glycation plays a significant role in age-related formation of cataracts, and it is possible that Hsp70 glycation could be used as a biomarker for cellular health in the future [100,101].

## 5. Influencing Chaperone Activity Via Pharmaceuticals

The mechanism of Hsp70 activity is defined by the ability of the chaperone to undergo large conformational changes that dictate binding to co-chaperones and client proteins. It was shown that subtle changes in lid positioning, based on different client bindings, could lead to changes in activity. A range of structures demonstrate that the SBD helical lid changes position when binding to large versus small proteins, unstructured polypeptides, or even amyloid fibrils [102]. The domain structure of Hsp70 is highly flexible, and subdomains reorient upon client binding [103]. Considering alteration of the Hsp70 structure can fine-tune its function, it is of no surprise that post-translational modifications would have a significant effect on chaperone function.

How can post-translational modification research on the Hsp70/Hsp40 system inform novel mechanisms to treat diseases of protein misfolding? We have discussed multiple examples of how PTMs can act as a molecular switch between pathways, alter binding between co-chaperones, alter binding to client proteins that can alter cellular signaling, and even increase affinity or rates of client refolding. If pharmaceutical interventions can be used to manipulate these events, it might be possible to influence overall chaperone function within a cell. Hsp90 plays a role in cancer and has been a target of potential mechanisms for chemotherapy, yet co-chaperones and PTMs alter the efficacy of these compounds [104]. It is plausible that there are compounds that will similarly affect Hsp70.

Currently, there are a host of pharmaceutical methods that are being investigated that may alter chaperone activity, with the intent on being developed for disease intervention. There are multiple compounds that bind to allosteric sites delaying the release of ADP from the nucleotide binding pocket, and this increases the amount of time a client protein interacts with Hsp70 [105]. Additionally, these compounds are effective in increasing Hsp70 binding to phosphor-tau, which in turn reduces tau polymerization into toxic aggregates in tissue culture models [106]. Additionally, it is possible to mimic Hsp40 binding via small molecules or biologically active peptides to stimulate Hsp70 activity, which leads to significantly increased clearance of poly-Q aggregates in a yeast model [107].

Similarly, by influencing the interactions between Hsp70 and HOP or CHIP, the balance between refolding and degradation can be shifted. This strategy is utilized with a compound that promotes HOP binding to Hsp90, wherein if HOP is forced to interact with Hsp90, it cannot interact with Hsp70. This enables CHIP binding and targeting proteins for degradation, which has proven effective for alpha-synuclein aggregate clearance [108]. Since PTMs mitigate the switch between HOP and CHIP, they make excellent targets for small molecule binding and potentially influencing this switch. This concept opens up the possibility that small molecules might be used to reduce apoptotic signaling, preserving neuronal health as neurodegenerative diseases progress. Additionally, mimicking acetylation could increase autophagic pathways to improve clearance of toxic protein aggregates. As we continue to study the structural changes caused by PTMs, we will discover new interactions and conformations that will more fully describe the complement of functions made possible by the Hsp70 chaperone system.

## Figures and Tables

**Figure 1 ijms-20-04207-f001:**
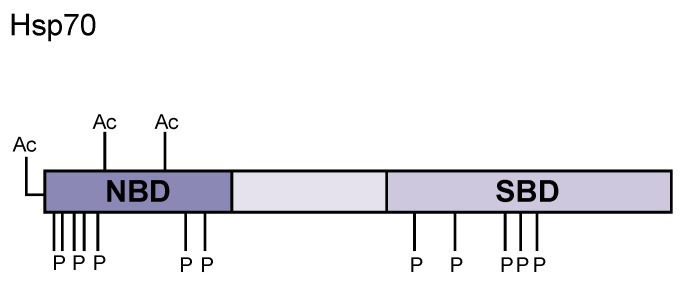
Hsp70 domain structure and post-translational modifications. Hsp70 contains an N-terminal nucleotide binding domain (NBD) that binds to ATP and catalyzes ATP hydrolysis upon client protein binding. The middle region consists of a charged, flexible linker region that enables a high degree of mobility for the C-terminus. The C-terminal domain is a bilobed substrate binding domain (SBD) where unfolded client proteins bind, inducing conformational change to bring the N- and C-terminal domains close together. Many post-translational modifications covered in this review are highlighted in this diagram (Ac = acetylation, P = phosphorylation).

**Figure 2 ijms-20-04207-f002:**
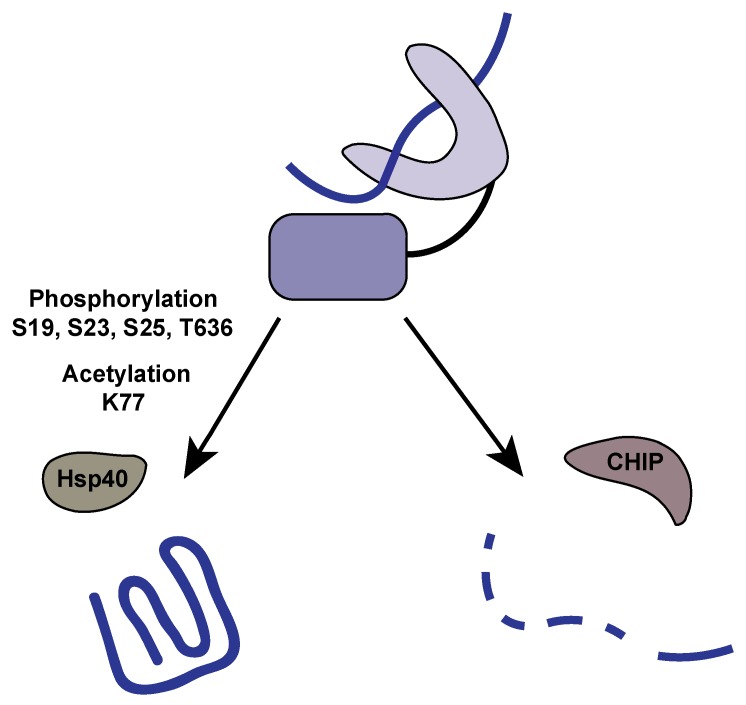
Post-translational modifications influence the fate of client proteins. The fate of client proteins bound to Hsp70 are determined by co-chaperone binding, which is influenced by post-translational modifications (PTMs). Phosphorylation and acetylation of Hsp70 promote binding to Hsp40 co-chaperones like HOP, which stimulates client protein refolding. Alternatively, when Hsp70 is dephosphorylated and deacetylated, CHIP binds and promotes ubiquitinylation and subsequent degradation of client proteins.

**Table 1 ijms-20-04207-t001:** Effects of post-translational modifications on chaperone function. Overview of major post-translational modifications and their effect on chaperone function as covered in this review.

Chaperone	Position	Modification	Effect
**Hsp70**	T13	Phosphorylation	Centrosome localization
	S19	Phosphorylation	Promotes protein refolding and aggregate binding
	S23	Phosphorylation	Promotes protein refolding and aggregate binding
	S25	Phosphorylation	Promotes protein refolding and aggregate binding
	T66	Phosphorylation	Kinetochore microtubule localization
	T226	Phosphorylation	Centrosome localization
	S362	Phosphorylation	Centrosome localization
	S486	Phosphorylation	Stabilize Caspase-3 and inhibit apoptosis
	T492	Phosphorylation	Promotes growth
	S495	Phosphorylation	Promotes growth
	T499	Phosphorylation	Promotes growth
	T504	Phosphorylation	Promotes dimer assembly
	S631	Phosphorylation	Centrosome localization
	T633	Phosphorylation	Centrosome localization
	T636	Phosphorylation	Promotes Hsp40 STi1/HOP binding
	N-term	Acetylation	Not specified
	K77	Acetylation	Promotes apoptosis
	K126	Acetylation	Promotes autophagy
**Hsp40**	S189	Sti1 (Hsp40)	Localized to nucleus
	T198	STi1 (Hsp40)	Localized to nucleus

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
