# Peer review of "Fine Tuning: Effects of Post-Translational Modification on Hsp70 Chaperones"

_ijms, 2019, doi:10.3390/ijms20174207_

Round 1

Reviewer 1 Report

The topic of Hsp70 family of chaperones is very broad and there is a multitude of reviews focusing on them. The review manuscript submitted for evaluation deals with just one aspect of these chaperones’ functions and properties, i.e. how posttranslational modifications modulate their functions within the cell. The review is logically divided into subsections starting with general introduction on posttranslational modifications of proteins and the functions and benefits of PTMs, then continuing with the description of the structure of Hsp70 class of proteins and their co-chaperones. In the following subsection the authors give the detailed account of two well known Hsp70 posttranslational modifications: phosphorylation and acetylation, their influence on Hsp70 activity and the importance of posttranslational modifications of Hsp70 for the regulation of Hsp70-dependent cellular processes. The authors also mention other less well studied modifications, such as ubiquitination, methylation, prenylation and glycation. In the last subsection the authors describe the influence of pharmaceuticals on Hsp70 class of proteins and its potential relevance to therapies of neurodegenerative diseases caused by protein misfolding or aggregation.

In this reviewer opinion the manuscript is well written and logically structured and will be informative for anyone interested in the functions of Hsp proteins and their regulation via posttranslational modifications.

This reviewer found only few minor errors:

line 79 - word “are” is duplicated

line 250 – word ”However” should not be capitalized

line 293 – archaic word “engulphed” could perhaps be substituted with modern “engulfed”

line 316 – instead of “occur” there should be “that occur” or “occurring”

line 322 – word “to” is missing

line 353 - word “Methylglyocal” should be substituted with “Methylglyoxal”

Author Response

We want to thank the reviewer for their thoughtful comments on the manuscript. 

We have fixed any of the following errors in the latest revision:

line 79 - word “are” is duplicated - deleted duplactaed word.

line 250 – word ”However” should not be capitalized - corrected capitolization. 

line 293 – archaic word “engulphed” could perhaps be substituted with modern “engulfed” - changed to read "engulfed"

line 316 – instead of “occur” there should be “that occur” or “occurring” - changed to read "that occor"

line 322 – word “to” is missing - added the requisite "to"

line 353 - word “Methylglyocal” should be substituted with “Methylglyoxal” - made the correct substituion. 

Thank you again for taking the time to review our manuscript.

Reviewer 2 Report

Dear Editor and authors

The paper is very interesting,  however

I suggest some editing to improve its value.

Two paragraphs need to be inserted, one regarding the methods for searching the literature and another about the relationship between chaperone and apoptosis. 

I also  suggest to add some missing bibliographic data to be use for comparison in the discussion  since the connection of Hsp and cell cycle. Future Science OA 2019;5(5) doi 10.2144/fsoa-2019-6017 regarding cell cycles regulators and smoke

Cells 2019;8(8), doi 10.339010.3390/cells 8080806 regarding the interaction with protein drivers

Future oncology 2013;9(5):649-55 about the perspective of target molecules in lung carcinogenesis.

Curr Mol Med 2018;18(6):343-51 for comparison of chaperone with HIF in cell cycle regulation

Nature Rev Mol cell biol 2019 about chaperone network

Cell Mole Life Sci 2005;62(6):670-84  about the regulation of kinases, apoptosis and APC activation by Hsp

Author Response

We would like to thank the reviewer for taking the time to review our manuscript. The comments were addressed in the revised edition of the maunscript.

-paragraph was added highlighting the relationship nbetween chaperones and apoptosis [lines 280-298]

-Paragraph was added at the end of manuscript regarding the methods for literature search

-We read and reviewed each of the references listed by the reviewer. We updated the manuscript and added citations or referenced these papers in the latest revision. 

Thank you again for the reviewers thoughtful comments on our manuscript.